# INTEGRATE: Model-based multi-omics data integration to characterize multi-level metabolic regulation

**Marzia Di Filippo**[1,2], **Dario Pescini**[1,2], **Bruno Giovanni Galuzzi**[2,3], **Marcella Bonanomi**[2,3], **Daniela Gaglio**[2,4], **Eleonora Mangano**[5], **Clarissa Consolandi**[5], **Lilia Alberghina**[2,3], **Marco Vanoni**[2,3], **Chiara Damiani**[2,3]*

**1** Department of Statistics and Quantitative Methods, University of Milan-Bicocca, Milan, Italy, **2** ISBE/SYSBIO Centre of Systems Biology, Milan, Italy, **3** Department of Biotechnology and Biosciences, University of Milan-Bicocca, Milan, Italy, **4** Institute of Molecular Bioimaging and Physiology (IBFM), National Research Council (CNR), Segrate, Italy, **5** Institute for Biomedical Technologies (ITB), National Research Council (CNR), Segrate, Italy

* chiara.damiani@unimib.it

**Data Availability Statement:** All relevant data are within the manuscript and its Supporting information files. Code to reproduce the overall

## Abstract

Metabolism is directly and indirectly fine-tuned by a complex web of interacting regulatory mechanisms that fall into two major classes. On the one hand, the expression level of the catalyzing enzyme sets the maximal theoretical flux level (i.e., the net rate of the reaction) for each enzyme-controlled reaction. On the other hand, metabolic regulation controls the metabolic flux through the interactions of metabolites (substrates, cofactors, allosteric modulators) with the responsible enzyme. High-throughput data, such as metabolomics and transcriptomics data, if analyzed separately, do not accurately characterize the hierarchical regulation of metabolism outlined above. They must be integrated to disassemble the interdependence between different regulatory layers controlling metabolism. To this aim, we propose INTEGRATE, a computational pipeline that integrates metabolomics and transcriptomics data, using constraint-based stoichiometric metabolic models as a scaffold. We compute differential reaction expression from transcriptomics data and use constraint-based modeling to predict if the differential expression of metabolic enzymes directly originates differences in metabolic fluxes. In parallel, we use metabolomics to predict how differences in substrate availability translate into differences in metabolic fluxes. We discriminate fluxes regulated at the metabolic and/or gene expression level by intersecting these two output datasets. We demonstrate the pipeline using a set of immortalized normal and cancer breast cell lines. In a clinical setting, knowing the regulatory level at which a given metabolic reaction is controlled will be valuable to inform targeted, truly personalized therapies in cancer patients.

workflow is available at: https://github.com/qLSLab/integrate.

**Funding:** LA received funding from Ministero dell'Istruzione, dell'Università e della Ricerca (MIUR), grant ITFoC MV received funds from Ministero dell'Istruzione, dell'Università e della Ricerca (MIUR), grant CHRONOS ('Dipartimenti di Eccellenza 2017') LA received funds from Ministero dell'Istruzione, dell'Università e della Ricerca (MIUR), grant 2020-NAZ-0057/A - JRU ISBE-IT - BTBS The funders had no role in study design, data collection and analysis, decision to publish, or preparation of the manuscript.

**Competing interests:** The authors have declared that no competing interests exist.

## Author summary

The study of metabolism and its regulation finds increasing application in various fields, including biotransformations, wellness, and health. Metabolism can be studied using post-genomic technologies. Transcriptomics, proteomics and metabolomics provide snapshots of transcripts, enzyme levels and metabolites in specific physio-pathological conditions. In the health field, the transcriptome and, more recently, the metabolome have been broadly profiled at the pre-clinical and clinical levels, while only more recently proteomic studies profiling metabolic enzymes are becoming available. However, the informative power of single omic technologies is inadequate since metabolism regulation involves a complex interplay of regulatory steps. Gene expression regulates metabolism by setting the upper level of metabolic enzymes, whereas the interaction of metabolites with metabolic enzymes directly auto-regulates metabolism. Therefore, there is a need for methods that integrate multiple data sources. We present INTEGRATE, a computational pipeline that captures dynamic features from the static snapshots provided by transcriptomics and metabolomics data. Through integration in a steady-state metabolic model, the pipeline predicts which reactions are controlled purely by metabolic control rather than by gene expression or a combination of the two. This knowledge is crucial in a clinical setting to develop personalized therapies in patients of multifactorial diseases, such as cancer. Besides cancer, INTEGRATE can be applied to different fields where metabolism plays a driving role.

## Introduction

Many physio-pathological states and multifactorial diseases, from cancer [1] to neurodegeneration [2] and aging [3] show a specific metabolic component. By its very nature, metabolism is closely integrated with most—if not all—cellular processes. For this reason, metabolism may act as a specific integrative readout of the physio-pathological state of a cell or organism [4, 5].

While the general topology of metabolism is well established, the characterization and understanding of system-level regulation of metabolism remain largely unresolved. However, some general rules have emerged in recent years [6, 7]. Each metabolic flux depends on at least two intertwined regulatory layers [8–10], as described below.

1. The upper level for the flux of each enzyme-catalyzed metabolic reaction depends on metabolic enzymes. The levels and catalytic activities of metabolic enzymes are set by complex, hierarchical regulatory mechanisms, orchestrated by signal transduction pathways [11]. The regulatory mechanisms include epigenetic control of chromatin and accessibility to transcription factors [12], the rate of transcription of the individual genes encoding metabolic enzymes [13], and post-transcriptional and post-translational events (from RNA splicing to enzyme phosphorylation [11]).

2. Auto-regulation occurs through the interactions of metabolites (substrates, cofactors, allosteric modulators) with the responsible enzymes. The flux of an enzyme-catalyzed reaction depends on the concentration of its substrate(s). The dependence of the reaction rate on substrate concentration is approximated by the Michaelis-Menten law, for non-allosteric enzymes. According to such law, the reaction rate increases with increasing substrate concentration, asymptotically reaching the maximum velocity $V_{max}$. The Michaelis constant $K_m$ describes the substrate concentration at which the reaction rate is equal to $V_{max}/2$. At low substrate concentrations (below $K_m$), there is a linear increase in the reaction rate with

increasing substrate concentration. As the concentration of substrate increases (far above $K_m$), E becomes closer and closer to being saturated with S, so that variations in S will have little if any effect on the reaction velocity. Metabolites within the same pathway or belonging to other cross-related biochemical pathways can also fine-tune each enzyme-catalyzed reaction through allosteric effects that effectively up- or down-modulate the ability of the enzyme to catalyze the reaction at a given substrate concentration. This metabolic regulatory layer contributes to regulating metabolism at the system level and may even fully account for metabolic rewiring.

Hence, differences in metabolic fluxes are only partially determined by variations in protein/gene expression. Let us take, for example, a specific, irreversible metabolic reaction $r$ ($S \rightarrow P$) catalyzed by enzyme $E$. A significant increase in the abundance of enzyme E, induced by increased gene expression, might originate—or not—a change of flux through its cognate reaction $r$. The following scenarios can be envisioned (also depicted in S1 Fig).

- Transcriptional control: variations in the flux through $r$ are mainly determined by variations in the abundance of E, induced by gene expression. This scenario can occur, for instance, when S is in large excess and the enzyme is saturated (S concentration much larger than $K_m$). In such a situation, variations in substrate concentration do not significantly affect the flux, provided that S remains in excess after the variation in its abundance and/or in the abundance of E.

- Metabolic control: variations in the flux through $r$ are mainly determined by variations in the abundance of S. This scenario typically occurs when E is in large excess (S concentration below or near $K_m$). In such a situation, the flux is not significantly affected by variations in the abundance of E induced by gene expression, provided that E remains in excess after the variation in its abundance and/or in the abundance of S.

- Combined metabolic and transcriptional control: variations in fluxes are determined by concerted variations in the abundance of S and E. This situation can occur, for instance, when E is in moderate excess over S (S concentration slightly above $k_m$) and a significant increase in the abundance of S saturates the enzyme. The increase in the abundance of S alone is sufficient to increase the flux. Yet, a concurrent increase in the enzyme level can raise it further.

Characterizing the landscape of metabolism and its regulation is of paramount importance in various fields, including health, wellness, and biotransformations [14]. The first requirement for this characterization is the knowledge of metabolic fluxes. However, direct determination of metabolic fluxes through the use of labeled substrates lags behind other omic technologies, such as metabolomics and transcriptomics, mainly due to technical difficulties [15], especially at the sub-cellular level [16]. On the contrary, transcriptomics (or proteomics) and metabolomics datasets are increasingly being collected in large cohorts but do not allow for accurate characterization of the regulatory mechanisms controlling metabolism unless opportunely integrated. More recently, parallel transcriptomics and metabolomics datasets started to appear [17–22]. Nevertheless, the integration of transcriptomics with metabolomics data has so far been generally limited to gene-metabolite correlation analysis or pathway enrichment analysis of genes and metabolites [23, 24]. Hence, there is a need for data science methods to integrate these heterogeneous omics data into the same computational framework, so to capture all the facets of the interdependence between metabolism and gene expression.

Constraint-based steady-state models represent a valuable framework to predict metabolic fluxes from the other high-throughput omics data [25, 26]. In particular, a plethora of methods have been conceived to integrate transcriptomics data into these kinds of models by relying on

Gene-Protein-Reaction associations (GPRs) encoded within them, as reviewed in [27–29]. Intracellular metabolomics data have also been indirectly integrated into constraint-based steady-state models in the form of constraints on fluxes [30–33], aiming at identifying the metabolic flux distribution better fitting the given data.

Nevertheless, using cross-sectional metabolomics data to predict fluxes requires *a priori* assumptions on the relationship between metabolites and fluxes. For example, `iReMet-flux` [30] assumes linearity between substrate abundance and flux for all reactions. Pandey et al. [31], assume instead that the variation in the abundance of a given metabolite must translate into a variation in the fluxes responsible for either its production or consumption. Our aim is instead to use metabolomics data to assess *a posteriori* whether a variation in a flux is consistent with a given assumption. Our novel hypothesis is that evidence for a monotonic relationship between variations in fluxes and variations in substrate abundances, and for a concurrent non-monotonic relationship between flux variation and enzyme abundance variation, indicates that a reaction is controlled metabolically.

Current model-based attempts to discern transcriptionally from metabolically controlled fluxes present some limitations [34, 35]. For example, Katzir et al. [35] do not directly use metabolomics data to infer reactions controlled at the metabolic level. They determine them by elimination, that is, by associating them with fluxes that are not regulated at the transcriptional, translational, nor post-translational level. On the contrary, the approach by Cakir et al. [34] is based on the concept of neighborhood in a graph. It does not distinguish reactions substrates from products, nor enzyme subunits from isoforms, and does not predict whether a reaction is up or down-regulated, but simply de-regulated.

We present in this paper the INTEGRATE (Model-based multi-omics data INTEGRAtion to characterize mulTi-level mEtabolic regulation) pipeline to accurately characterize the landscape of metabolic regulation in different biological samples, starting from metabolomics and transcriptomics data.

INTEGRATE first computes differential expression of reactions from transcriptomics data (transcriptional regulation only).

Then, INTEGRATE exploits constraint-based modeling to predict how the global relative differences in expression are expected to translate into consistent differences in metabolic fluxes. To this aim, any available method can be used in principle. In this work, we took inspiration from the plethora of methods proposed in literature [27–29]. In particular, we set flux boundaries as a function of gene expression as done, among others, by eFlux [36] and TRFBA [37]. We used relative gene-expression values as in GX-FBA [26]. We scaled metabolic fluxes relative to the maximum flux identified using Flux Variability Analysis, as in scFBA [38]. To improve model predictions, as recommended in [29], INTEGRATE sets constraints also on selected extracellular fluxes, according to exo-metabolomics data.

In parallel, INTEGRATE uses intracellular metabolomics datasets and the mass action law formulation to predict how differences in substrate availability translate into differences in metabolic fluxes (metabolic regulation only), neglecting enzymatic activity. The intersection of the two output datasets discriminates fluxes regulated at the metabolic and/or gene expression level.

As a proof-of-principle of the pipeline, we generated transcriptomics and metabolomics data for a set of immortalized normal and cancer breast cell lines. Using a newly developed, streamlined, manually curated, multi-compartment metabolic network of human central carbon metabolism, INTEGRATE proved able to identify statistically significant co-variations between experimental metabolomics and computationally predicted fluxomics. Knowing the regulation layer altered in a specific patient may prove useful in the development of truly personalized therapies.

## Results

### The INTEGRATE pipeline

We conceived the INTEGRATE methodology to particularize the hierarchical regulation of metabolic differences across different groups of biological samples. For the sake of readability, we refer to different groups of samples simply as cell lines, which represent the application scope of this work, bearing in mind that the methodology is general and can be applied to any group of samples, such as tissues from different patients.

INTEGRATE takes as input 1) a generic metabolic network model, including GPRs 2) cross-sectional transcriptomics data 3) cross-sectional intracellular metabolomics data 4) steady-state extracellular fluxes data. INTEGRATE returns as final main output two lists of metabolic fluxes: 1) fluxes that vary across cells consistently with both metabolic and transcriptional regulation 2) fluxes that vary consistently with metabolic regulation only.

The core process of INTEGRATE methodology is depicted in Fig 1. It consists in integrating the input experimental datasets, which are centered around heterogeneous objects (i.e.,

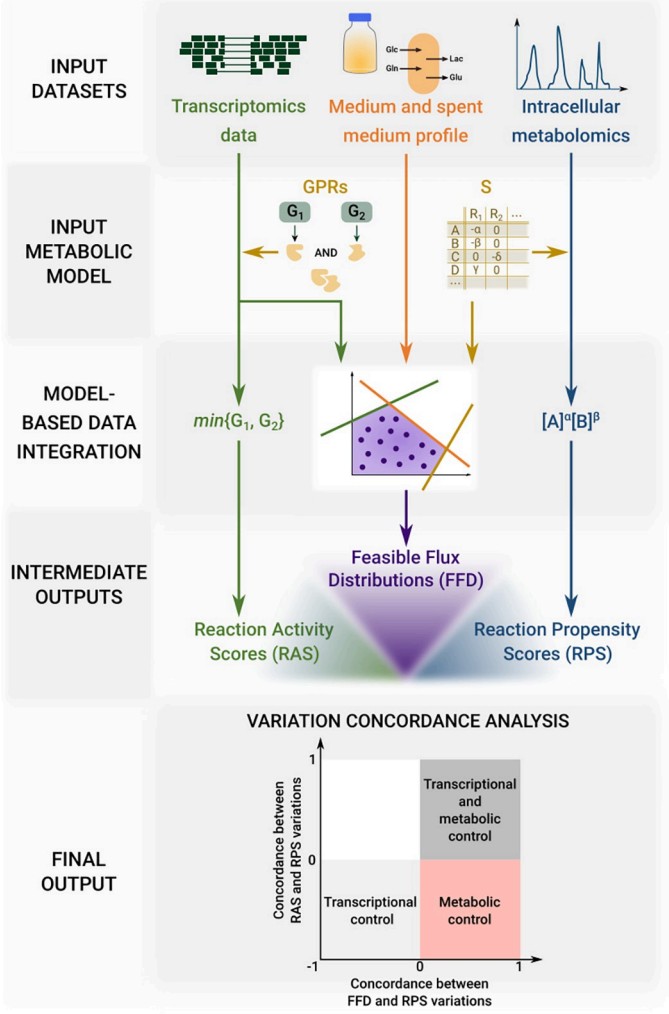

**Fig 1. Graphical representation of INTEGRATE pipeline.** Three heterogeneous experimental datasets are integrated into the input metabolic model to obtain three comparable datasets regarding transcriptomics, fluxomics and metabolomics. The concordance level between pairs of these intermediate output datasets is the final output of INTEGRATE.

genes, metabolites, and fluxes) into the input metabolic network to obtain the three following datasets, each of which is centered around the object reaction:

*Reaction Activity Scores (RAS).* This dataset includes a RAS score for each input model reaction and for each sample. The score is based on the expression value (RNA-seq read counts) of the genes encoding for catalyzing enzymes and on the relationship among them, as previously done in [39].

*Feasible Flux Distributions (FFD).* This dataset includes a large number of flux distributions associated with a given cell line, obtained by uniformly sampling the feasible flux region of the metabolic model. The model must have been previously tailored on the cell line by integrating transcriptomics and available information on extracellular constraints in the form of differential constraints relative to the other cell lines.

*Reaction Propensity scores (RPS).* This dataset includes an RPS score, based on the availability of reaction substrates, for (ideally) each input model reaction and for each sample. The score is computed as the product of the concentrations of the reacting substances, with each concentration raised to a power equal to its stoichiometric coefficient. According to the mass action law, the rate of any chemical reaction is indeed proportional to this product. This assumption holds as long as the substrate is in significant excess over the enzyme constant $K_M$. If one single reaction substrate is missing from the metabolomics measurements, the reaction is omitted from the dataset.

Once the three reaction-oriented datasets are obtained, INTEGRATE assesses for each of them whether the value of each reaction is significantly higher or lower in a given cell line as compared to another one. We consider a variation as statistically significant if both the null hypothesis is rejected according to any suited statistical test and if the variation exceeds a threshold value. See Material and methods for the specific tests that we used in this study.

INTEGRATE then assigns two scores to metabolic reactions. The first score quantifies the concordance level between the variation signs obtained for the RAS dataset and those obtained for the RPS dataset (for reactions in common). Highly concordant reactions correspond to fluxes whose metabolic and transcriptomic regulation is concerted, poorly concordant vice versa. The second score assesses the concordance between FFD and RPS (for reactions in common) and thus whether flux variations are consistent with metabolic regulation. Reactions with a low RAS-RPS agreement but a high FFD-RPS correspond to *metabolically controlled* reactions.

We remark that we preferred not to give the same attention to flux variations that are consistent with transcriptional regulation only, based on the concordance between RAS and FFD, because the two datasets are not independent.

Scripts to reproduce the overall workflow are available at https://github.com/qLSLab/integrate, and are also available in Zenodo under permanent identifier 10.5281/zenodo.5824504 (https://zenodo.org/badge/latestdoi/352613136).

## Selected breast cancer cell lines display heterogeneous metabolic profiles at balanced growth

We applied INTEGRATE to cell lines that we expected to be metabolically heterogeneous to test our approach. We selected four breast cancer cell lines, deriving from either primary or metastatic breast cancer tissues with different molecular classifications, and a non-tumorigenic

**Table 1. Characterization of the investigated non-tumorigenic and cancer breast cell line in terms of their subtype and origin.** ER stands for estrogen receptor, PR stands for progesterone receptor, whereas HER2 stands for human epidermal growth factor receptor-2. Cell lines may be positive (plus sign) or negative (minus sign) for each of the described subtypes. Breast cancer also includes Luminal A and B subtypes.

| Cell line | Subtype | | | | Origin |
|---|---|---|---|---|---|
| | ER | PR | HER2 | Luminal | |
| MCF102A | | | | | Non-tumorigenic breast epithelial cell line |
| SKBR3 | - | - | + | | Breast adenocarcinoma |
| MCF7 | + | + | - | A | Pleural effusion metastasis of a breast adenocarcinoma |
| MDAMB231 | - | - | - | | Pleural effusion metastasis of a breast adenocarcinoma |
| MDAMB361 | + | + | + | B | Brain metastasis of breast adenocarcinoma |

breast cell line. The name, origin, and molecular subtyping of the five cell lines are summarized in Table 1.

The five cell lines were cultured in a similar growth medium. It can be observed in Fig 2A and 2B that the cell lines present significant differences in terms of proliferation rate and protein accumulation.

Between 0 and 48 hours, the protein content linearly correlates with the number of cells (Fig 2C). Hence, biomass accumulation and cell division are balanced and cell size is constant in this time window. Therefore, we concentrated further analyses on the 0–48 hours interval.

We estimated the consumption and production rates of lactate, glutamine, glucose, and glutamate in the 0–48 hours time interval from YSI analysis of spent medium. We focused on these metabolites because glucose and glutamine are the primary carbon sources of cancer cells and because correct prediction of the rate of lactate and glutamate production over glucose and glutamine, by constraint-based models, is notoriously challenging [40, 41]. We quantified the abundance of intracellular metabolites and prepared libraries for RNA-sequencing at 48h.

It can be observed in Fig 2E that the cell lines present significant differences in terms of metabolic profile. The clusters of samples referring to each cell are indeed well separated from one another. In particular, as it can be observed in the dot plot in Fig 2D, the abundance of some metabolites well distinguishes a cell from the others. MCF7 cell line shows a prevalence of NADH involved in redox balance and fumaric acid, which is involved in the tricarboxylic acid cycle. MCF102A is enriched with D-Erythose 4-phosphate, which is involved in the pentose phosphate pathway. MDAMB231 cell line results enriched in metabolites involved in methionine and cysteine metabolism (Adenosine) and nucleotide synthesis (Adenine). MDAMB361 cell line shows enrichment in metabolites involved in glycolysis (D-glucose and Glyceric acid 1,3-biphosphate), methionine metabolism (L-homocysteine), and urea cycle (ornithine and citrulline). Finally, the most abundant metabolites in the SKBR3 cell line are involved in glycolysis (D-glucose and 3-Phosphoglyceric acid), cholesterol synthesis (Acetoacetic acid), and redox balance (NADPH).

Measurements on spent medium in Fig 2F show that the ratio of lactate produced over glucose consumed is quite similar across the five cell lines. In contrast, the lactate and glutamate over glutamine ratios are more heterogeneous.

All raw experimental data are provided in S1 File.

## The ENGRO2 metabolic model

Genome-wide reconstructions of human metabolism, such as Recon3D [42], are precious repositories of detailed and multi-level information about human metabolism. They involve

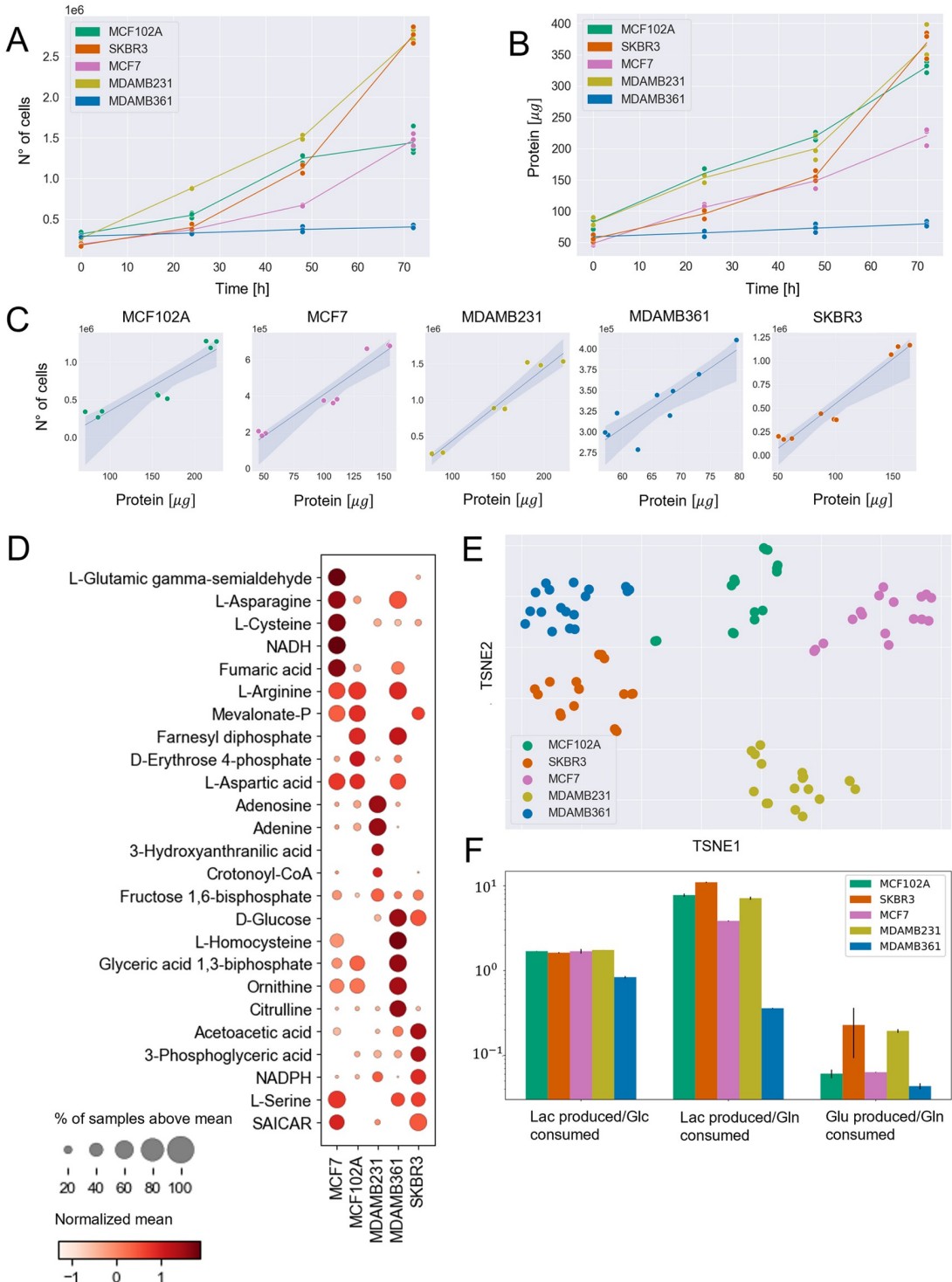

**Fig 2. Experimental metabolic measurements at balanced growth phase.** A) Number of cells in time. B) Protein content in time. C) Correlation plots between protein content and number of cells for experimental observations in the 0–48hrs time windows. D) Dot plot representing the normalized mean (Z-score) of the metabolite abundances within each line (visualized by color) and the fraction of samples of each cell line with the abundance over the mean of all cell lines samples (visualized by the size of the dot). The first five metabolites that better distinguish each cell line from the others, according to t-test p-value, are reported. E) t-SNE dimensionality reduction of intracellular metabolomics profiles. F) Extracellular flux ratios, derived from spent medium measurements.

thousands of metabolites, reactions, and genes. In principle, they can be used directly in our pipeline as a scaffold model for integrating the experimental input data relative to the five breast cell lines. However, in their current form, their comprehensiveness comes at the cost of some simulation issues and difficulties in interpreting the outcomes. The most relevant simulation problem is the presence of thermodynamically infeasible loops, which make the simulated growth rate insensitive to variation in essential nutrient availability constraints, as reported in [43]. For this reason, we preferred to use mainly a core model, extracted from Recon3D, focusing on more limited aspects of metabolism, but that underwent extensive manual curation and debugging.

To this aim, we reconstructed the ENGRO2 metabolic network, which is a constraint-based core model about central carbon metabolism and essential amino acids metabolism. ENGRO2 is a follow-up of the core model of human central metabolism ENGRO1 introduced in [44] to gain new knowledge about the logic of metabolic reprogramming in promoting cell proliferation under different nutritional conditions.

The reconstructed model has been refined by checking: i) the capability to reproduce ENGRO1 results [44] in terms of contribution of glucose and glutamine as carbon and nitrogen sources for supporting proliferative wirings and of sensitivity of the model at high and low levels of these nutrients; ii) the actual essentiality of essential amino acids, that is, null growth rate if they are depleted from the medium; iii) the capability to reproduce experiments in literature, including the dependence of cancer phenotypes from the *de novo* synthesis of palmitate-derived lipids rather than on an external source of fatty acids, as came out in [45], the *in silico* simulation of the effect of an inverse agonist for the nuclear receptor liver-X-receptor, whose role is to regulate the expression of some key genes in the glycolysis and lipogenesis, as a putative cancer treatment approach [46], the role of the creatine kinase (CK) enzyme that due to the requested high amount of ATP may act as potential anticancer agent [47], and the overexpression of argininosuccinate synthase enzyme as a mechanism for impairing cancer cells proliferation due to aspartate deviation from the production of pyrimidines [48].

The final version of the ENGRO2 core model consists of 494 reactions, 410 metabolites and 494 genes. As a comparison, we report that these values are respectively 84, 67, 216 for ENGRO1, and 13543, 8399, 3697 for Recon3D. A graphical representation of the network, split into two figures for improved readability, is reported in S2 and S3 Figs, depicting, respectively, the central carbon metabolism and the metabolism of the essential amino acids. The model is provided as SBML in S2 File and as XLSX in S3 File.

## Cell-relative metabolic models

We customized the ENGRO2 core model to obtain five cell-specific core constraint-based models of the cell lines under study. Because the cell-specific models must be functional to highlight the metabolic differences between the cell lines, we incorporated most constraints in the form of relative constraints. For this reason, the models cannot be considered as cell-specific stand-alone models. Hence we refer to them as *cell-relative* models.

Specifically, we integrated the following three kinds of relative constraints:

1. *Constraints on nutrient availability*. The network can internalize only metabolites that are supplied in the medium. These constraints also reflect the slight differences among the growth medium of the five cells. Similarly to what was done in [49], if the concentration of metabolite X in the medium of cell A is, for instance, 20% higher than in the medium of Cell B, the maximum uptake flux allowed for metabolite X in cell A will be 20% greater than that of cell B.

2. *Constraints on extracellular fluxes*. We set constraints on the ratios of the metabolites for which we have estimated the consumption and production rate, namely on the glutamate/ glutamine, lactate/glucose and lactate/glutamine ratios. The choice of constraining the relative ratios among these metabolites rather than their absolute intake or secretion rates is motivated by the limited subset of measured metabolites and the need to avoid an imbalance between such values and the arbitrary absolute values of type 1 constraints. In this way, the relative ratios between boundaries on extracellular fluxes are preserved both within and across cells.

3. *Transcriptomics-derived constraints on internal fluxes*. Each reaction is assigned a Reaction Activity Score (RAS), according to the expression of its associated genes and on the relationship among them encoded within GPR associations. The differences in the flux boundaries of any given reaction across the five cell lines reflect the differences in their RAS. More in detail, the cell line with the highest RAS can reach the maximum possible flux value, which is determined by type 1 and type 2 constraints. In contrast, the other cell lines have a RAS-proportionally reduced flux capability.

To verify that the integration of the above constraints segregates the feasible flux distributions of each cell line, we extensively sampled the feasible region of each model in the scenarios in which each of the three constraint is either applied alone or in combination with the others. We then applied a t-distributed stochastic neighbor embedding (t-SNE) algorithm and represented in Fig 3 the high-dimensional sampled flux distributions in a two-dimensional space [50]. It is possible to appreciate how the simultaneous application of the three constraints (Fig 3D) better separates the flux distributions sampled from each model (corresponding to the specific colour in the plots) from one another. Notably, constraints on extracellular fluxes alone (Fig 3B) do not allow the feasible flux distributions of the five models to be discriminated. On the contrary, transcriptomics-derived constraints alone (Fig 3C) result in a good separation of the feasible regions of the five models. The combination of both kinds of constraints (Fig 3D) decreases the separation between the sampled solutions for a given model (intra-model) and increase inter-model separation.

Similar conclusions are derived when considering the correlation between the growth yield on glucose computed starting from experimental and computational data (on average over the sampled FFDs). The experimental growth yield (reported in S1 File) was computed for each of the two collected biological replicates as the ratio of the total proteins produced over 48 hours (grams/hour, determined by Bradford analysis) over the glucose consumed over the same period (grams/liter, obtained by YSI analysis of spent medium. The counterpart *in silico* growth yield was computed as the median ratio of the protein synthesis flux over the glucose uptake flux. Both fluxes were expressed as grams consumed or produced/hour. The condition where transcriptomics-derived constraints alone are integrated well discriminates the five cell lines in terms of their growth rate, as shown in the relative correlation plot in Fig 3. However, predictions of growth rates improve when constraints on extracellular fluxes are also added.

The final five cell-relative metabolic models (SBML format) are included in S1 Compressed File Archive. It must be specified that the SBML format does not embed type 2 constraints (constraints on extracellular fluxes) which must be specified separately.

## INTEGRATE discriminates reactions regulated at different levels

By integrating the information of the three derived datasets, we could ascertain at which level each reaction is controlled. We measured the qualitative concordance between the RAS and RPS, as well as between RPS and FFD values in all pairwise comparisons between cell lines, for

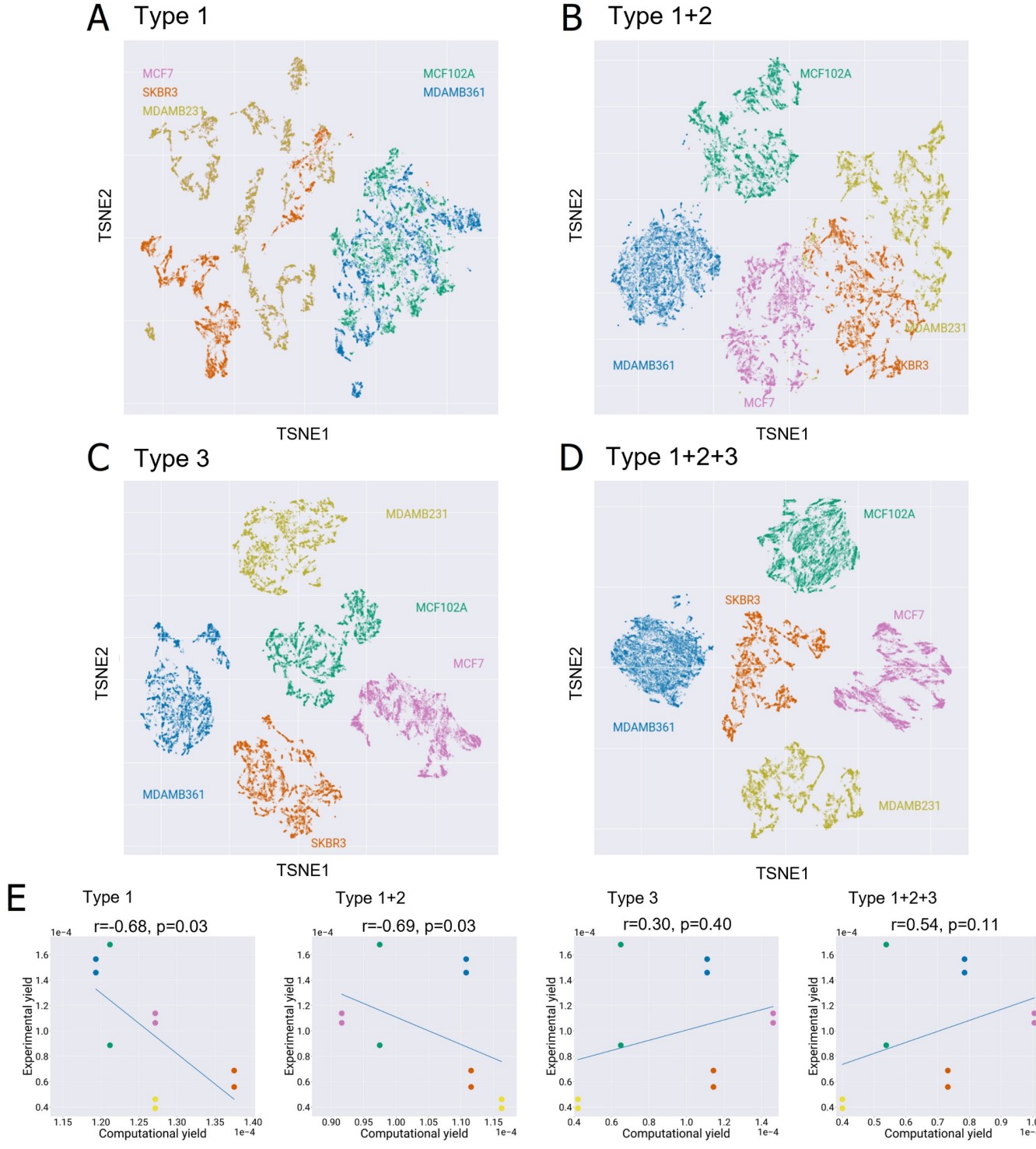

**Fig 3. Evaluation of the effect of the different types of constraints on ENGRO2 feasible solutions.** Effect of constraints on A) nutrients availability (type 1), B) nutrients availability and extracellular fluxes (type 1+2), C) intracellular fluxes based on transcriptomics data (type 3) and all together (type 1+2+3) in segregating the five investigated cell lines. A two-dimensional map of the FFDs of the five cell lines in each setting is shown. For reversible reactions, that net flux is considered. For computational reasons, only 10000 steady-state solutions sampled within the feasible region of each model were plotted. E) Correlation between the experimental and *in silico* growth yield on glucose is reported for each of the four settings in panels A, B, C and D. The Spearman correlation coefficient and p-value are reported on top of each plot.

all eligible reactions, i.e., reactions for which metabolite levels for all substrates are available. We focused on the qualitative concordance, that is, the concordance of variation sign, rather than on a quantitative concordance of numerical variations values, as we cannot expect proportionality between RPS and FFD, nor between RAS and RPS values.

Qualitative concordance was measured by the Cohen's kappa coefficient, which quantifies the difference between the rate of agreement that is actually observed and the rate of agreement that would be expected purely by chance. The value of Cohen's kappa is 1 if the two datasets are fully concordant; 0 if they agreed only as often as they would by chance. A negative value of Cohen's kappa indicates that the two datasets agreed even less often than they would by chance. A value of –1 means that the two raters made opposite judgments in every case. This metric allowed us to rank reactions according to their concordance. We remark that Cohen's kappa is not a statistical test that provides a well-defined yes/no result. However, it has been recommended [51] to consider a value below 0.2 as poor concordance, a value between 0.21 and 0.40 as fair, between 0.41 and 0.60 as moderate, between 0.61 and 0.80 as good, and between 0.81 and 1.0 as very good agreement.

As a general overview on the INTEGRATE outcomes, Fig 4A reports the concordance level between RAS and RPS variations (briefly RPSvsRAS) versus the concordance level between RPS and FFD variation (briefly RPSvsFFD), for the 81 metabolic reactions of ENGRO2 for which quantification of all substrate abundances was available. It can be observed that reactions distribute among the following categories:

- Reactions displaying positive values for both RPSvsFFD and RPSvsRAS scores (first quadrant—gray shadow in Fig 4A). Variations in these reactions must be imputed to transcriptional and metabolic regulation.

- Reactions having positive values for RPSvsFFD and negative RPSvsRAS scores (fourth quadrant—pink shadow in Fig 4A). Variations in these reactions must be imputed to metabolic control only.

- Reactions having negative values for both RPSvsFFD and RPSvsRAS scores, but high values for RASvsFFD concordance (third quadrant—white shadow in Fig 4A). Variations in these reactions must be imputed to transcriptional control only.

- Reactions having positive values for RPSvsRAS but negative RPSvsFFD scores (second quadrant in Fig 4A). The phenomenon according to which gene expression and substrate availability agree with one another, but the flux does not agree with any of the two, might be imputed to model inaccuracy, as well as to allosteric regulation, product inhibition, or cofactors/prosthetic groups. To properly discriminate among these scenarios, other kinds of omics data would be necessary. Yet, in our case study, only a few reactions belong to this category and with statistically not significant concordance levels. Hence, we simply labeled this category as 'other'.

The heatmap in Fig 4B reports the RPSvsRAS and the RPSvsFFD concordance scores (Cohen's kappa) of ENGRO2 metabolic reactions, limited to the subset (of cardinality 81) of reactions for which quantification of all substrate abundances was available. The values are ranked according to RPSvsFFD concordance scores. Only reactions with a RPSvsFFD concordance score higher than 0.2. Remaining reactions are reported in S4 File.

It can be observed that 4 reactions resulted from consistent transcriptional and metabolic regulation.

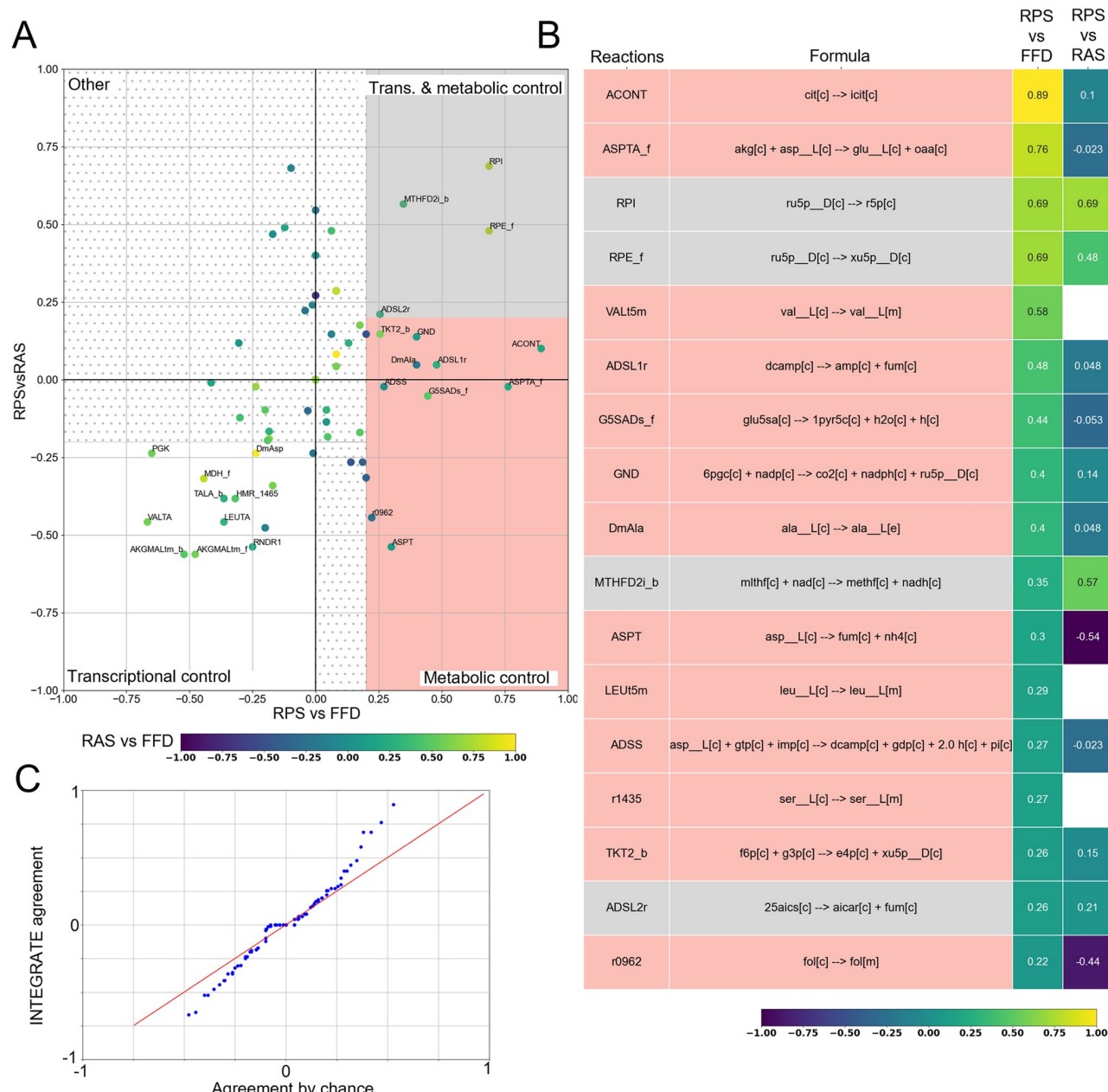

**Fig 4. Variation concordance analysis.** A) RPSvsFFD (x-axis) and the RPSvsRAS (y-axis) scores of the 81 metabolic reactions for which quantification of all substrate abundances was available. The points are coloured as a function the RASvsFFD scores. We reported the names of the reactions having at least one of the scores greater than 0.2 (i.e. fair concordance). B) Heatmap showing the RPSvsRAS and the RPSvsFFD concordance scores, for reactions having a level of concordance between RPS and FFD greater than 0.2. C) $Q - Q$ plot between the empirical probability of agreement between two independent datasets and INTEGRATE Cohen's kappa distribution related to the comparison between RPS and FFD.

On the contrary, 13 reactions resulted only metabolically regulated because of a RPSvsFFD score above 0.2 and a RPSvsRAS score below this threshold or even missing. Missing RPSvsRAS values occur when a reaction is not associated with a GPR.

In Fig 5A the mean RPS (on the left) and the FFD (on the right) of the previously identified reactions resulting from a consistent transcriptional and metabolic regulation or only

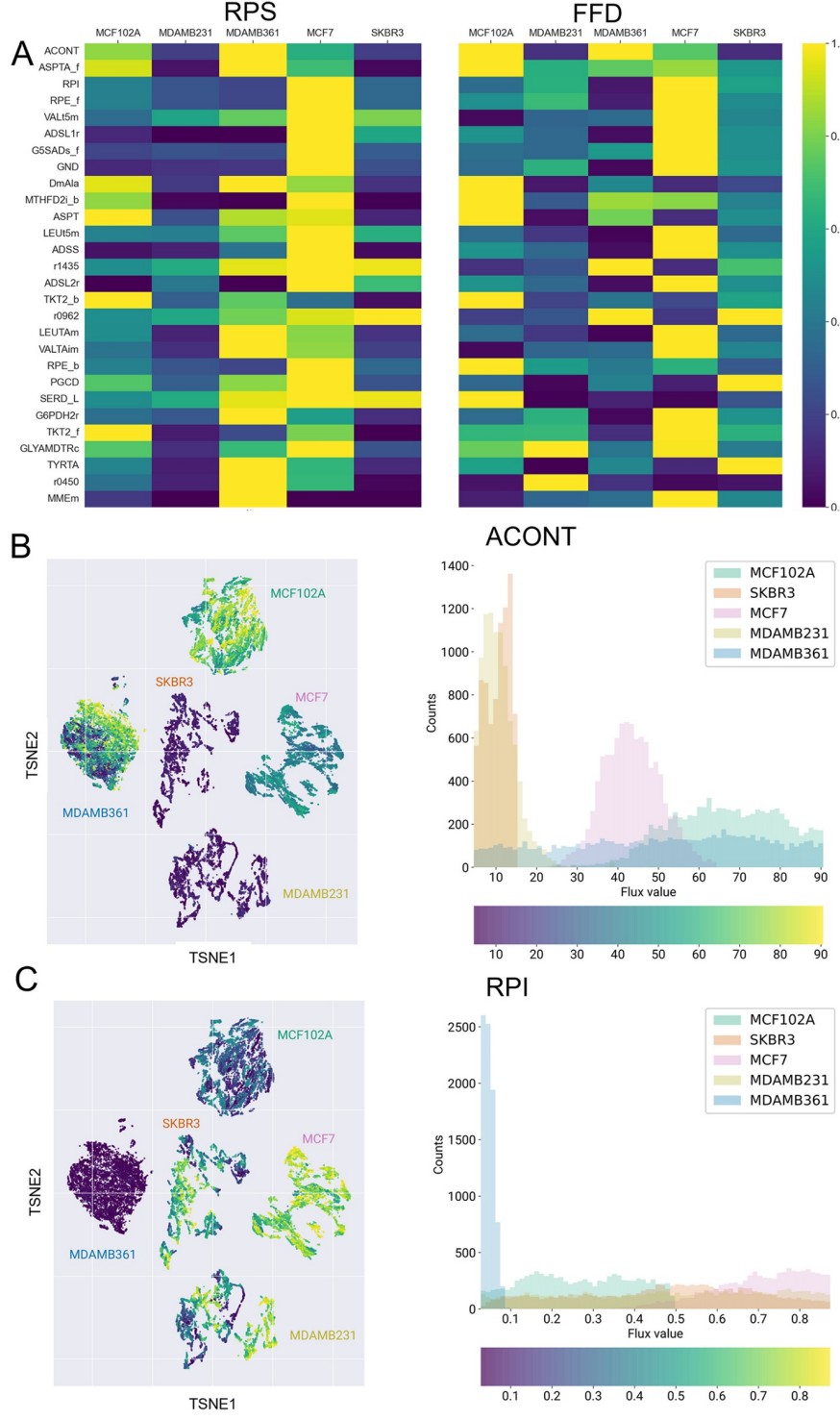

**Fig 5.** A) Normalized average RPS and median FFD for reactions in Fig 4B. B) Left: same as Fig 3A (constraints 1+2 +3) but with dots coloring representing the flux of cytosolic ACONT. Right: distribution of cytosolic ACONT flux values within the five cell lines. Histogram colors represent cell lines labels. C) Same as B for RPI reaction.

metabolically controlled are shown. For each reaction, the values are normalized by dividing them by the highest one. The maps and histograms in Fig 5B and 5C report respectively the FFD values distribution within the five cell lines of the most significant metabolically controlled reaction—namely the ACONT reaction, which catalyzes the production of cytosolic isocitrate from citrate—and metabolically and transcriptionally controlled reacton—namely the RPI reaction, which catalyzes the conversion of ribulose 5-phosphate to ribose 5-phosphate within the non-oxidative phase of the pentose phosphate pathway.

## Robustness of INTEGRATE results

We verified that the obtained concordance scores are robust to the choice of the sample. In fact, the standard deviation of the Cohen's kappa values across the ten batches of the total sample is negligible for most reactions, as reported in S4 File.

Although Cohen's kappa takes into account the possibility of the agreement occurring by chance, when assessing Cohen's kappa of multiple reactions, the probability of observing a high Cohen's kappa value by chance is expected to increase.

To counteract this problem, at first instance, we obtained the empirical probability of agreement between two independent datasets (see Material and methods). The comparison in Fig 4C between the distribution of the concordance values obtained with INTEGRATE and the empirical probability of agreement between two independent datasets confirms that a few fluxes agree (or disagree) with RPS variations beyond a reasonable doubt. On the contrary, for the majority of flux variations, the agreement is not significant. The non-significance can indicate that FFDs and RPSs do not agree, as well as that the agreement is uncertain. This uncertainty mainly depends on the limited number of cell lines for which we are assessing the agreement. Other possible sources of uncertainty are the uncertainty in the experimental data, the strictness of the mass balance constraint, and the limited scope of the ENGRO2 metabolic network.

To deal with the multiple comparisons problem, we then associated an empirical p-value to our obtained Cohen's kappa values and adjusted it with the Benjamini and Hochberg procedure to keep the False Discovery Rate (FDR) below 5%. The empirical and adjusted p-values are included in S4 File.

After FDR correction of the p-values, the concordance between RPSs and FFDs resulted statistically significant for reactions ACONT, ASPTA (forward), RPI, and RPE. It is however important to mention that FDR correction is a standard practice in gene expression analyses, but it might be too stringent in our case. In particular, one should take into account the lower probability of simultaneously observing a high RPSvsFFD concordance and a high RPSvsRAS concordance by chance.

To roughly assess the sensitivity of INTEGRATE results to the input metabolic model, we re-executed the overall pipeline on Recon3D [42], by limiting the model to the set of exchange reactions that are active in ENGRO2. Fig 6 reports the average concordance score obtained for 10 batches of 5000 points each. First of all, it is important to stress that concordance values are, as expected, highly sensitive to the sample size in the genome-wide model (S5 File). To avoid undersampling problems, a larger sample is needed, which is computationally more demanding as compared to a core model. Yet, it can be observed that, also in Recon3D, reactions distribute across the three categories.

Moreover, the comparison of the distribution of the concordance scores (Fig 6C) against the empirical probability distribution of the concordance between independent datasets resembles the ENGRO2 case (Fig 4C), confirming that in both models a few flux variations agree with RPS variations beyond a reasonable doubt.

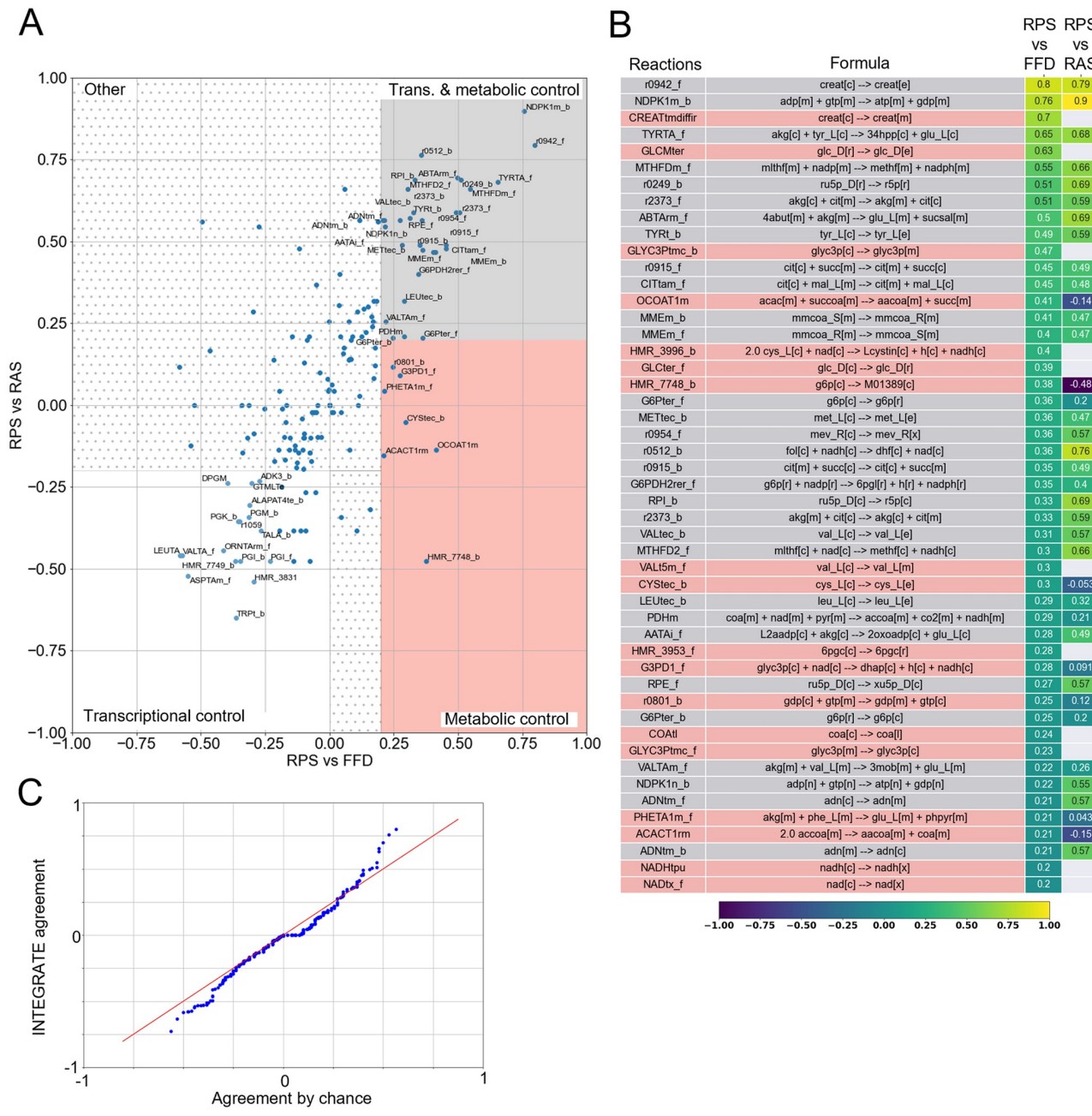

**Fig 6. Variation concordance analysis for Recon3D.** A) RPSvsFFD (x-axis) and the RPSvsRAS (y-axis) scores of the metabolic reactions for which quantification of all substrate abundances was available. We reported the names of the reactions having at least one of the scores greater than 0.2 (i.e. fair concordance). B) Heatmap showing the RPSvsRAS and the RPSvsFFD concordance scores, for reactions having a level of concordance between RPS and FFD greater than 0.2. C) $Q-Q$ plot between the empirical probability of agreement between two independent datasets and INTEGRATE Cohen's kappa distribution related to the comparison between RPS and FFD.

As expected the sets of reactions identified by the two models do not coincide. Yet some overlaps can be observed. For example, the RPE reaction, which is involved in the pentose phosphate pathway, appears to be transcriptionally and metabolically regulated in both models.

## Discussion and conclusion

We presented the INTEGRATE computational framework, that integrates transcriptomics and metabolomics data with stoichiometric models to unravel whether specific metabolic fluxes are governed by transcriptional regulation, differences in substrate levels, or both.

INTEGRATE first uses transcriptomics data—translated to Reaction Activity Scores (RASs) through GPR associations—and available information on extracellular fluxes to reconstruct metabolic fluxes, by constraining simulations of the input metabolic model. In parallel, it exploits intracellular metabolomics data—translated to Reaction Propensity Scores (RPSs) through reactions stoichiometry—to reconstruct how differences in substrate availability should translate into differences in metabolic fluxes if the role of variations in enzymatic activity was neglected and if substrate concentration was below the Michaelis-Menten constant. Then, without the ambition to be exhaustive, INTEGRATE classifies as metabolically regulated reactions that show a monotonic relationship between RPS and flux and as transcriptionally regulated reactions displaying a monotonic relationship between RAS and flux. Finally, INTEGRATE identifies fluxes that vary consistently at the metabolic and transcriptional regulation level and fluxes whose variation is concordant with metabolic regulation only.

To test our pipeline, we used both a manually curated, simulation-ready reconstruction of the human central carbon metabolism and, in parallel, the latest genome-wide reconstruction of human metabolism Recon3D. Application of our pipeline to one non-tumorigenic and four different breast cancer allowed us to ascribe some flux variations to either transcriptional or metabolic control or both. Remarkably, we identified reactions for which there is a good agreement between flux variations and variations in RPSs in both models. We remind that the two datasets are fully independent.

INTEGRATE captures dynamic features of the metabolic state of different cells or tissues from the integration of high-throughput data that provide complementary views on their static profile. The model-based integration of transcriptomics and metabolomics data enriches their expressive power. Metabolic fluxes predicted by constraint-based modeling complement information on the differential activity of reactions derived from gene expression data, with the information on the direction of the observed variation.

INTEGRATE also complements information on the differential propensity of reactions derived from metabolomics, with information on the compartment in which the reaction occurs. For instance, metabolomics data alone would not have allowed us to differentiate between the contribution of the aconitase reaction (ACONT) substrate when located in the cytosol compartment or when coming from its metabolic counterpart. On the contrary, INTEGRATE indicated that the metabolic regulation involves the cytosolic reaction. Notably, this information was complemented by transcriptomics data even though the aconitase flux itself is not regulated at the transcriptional level. This result indicates that indirect transcriptional regulation is likely to be responsible for observed differences in this flux, which would deserve further investigation.

Many analyses can be conceived and performed downstream of our pipeline. As an example, one may investigate the metabolism of amino acids, to analyze whether they tend to be preferentially synthesized or metabolized by each cell line. One may also want to analyze fluxes that better discriminate the five cell lines.

The main limitation of INTEGRATE is the usage of transcriptomics data to infer reaction fluxes, which as highlighted in [28] is prone to prediction errors. Yet, INTEGRATE uses some precautions to mitigate these errors. First of all, it does not have the ambition to predict exact flux values, but rather the sign of the variation across the cell lines. Moreover, it uses transcriptomics data on top of constraints on extracellular fluxes and growth. Constraints on

extracellular fluxes and metabolic growth, along with the mass balance constraint, relax the assumption on the linear relationship between gene expression and flux boundaries. Another limitation of constraint-based modeling, in general, is the sensitivity of results to the choice of the metabolic network. Indeed, we identified statistically significant concordances between fluxes and metabolomics in the core model, which are not significant in Recon3D and vice versa. Ad hoc experiments are thus required to validate model results and our underlying hypothesis that the agreement between fluxes and RPSs, and between fluxes and RASs, can be used to distinguish whether a reaction is controlled by transcriptional regulation or by substrate levels. In the meantime, we think that our proposal can be fuel for thoughts in the COBRA and cancer metabolism community.

The direct exploitation of metabolomics data to determine whether a flux is regulated at the metabolic level represents the main novelty of our approach. At first instance, we intercepted this information with information on transcriptional regulation. Nevertheless, the pipeline can be promptly extended to integrate proteomics and phosphoproteomics. One might indeed compute (phospho)proteomics-derived RASs, along with transcriptomics-derived RAS. Proteomics-derived RASs are expected to reflect enzymatic activity better than transcriptomics-derived ones, yet the latter offers better coverage. For this reason, one should still use transcriptomics-derived RAS to constrain the metabolic fluxes of reactions for which quantification of any protein involved in its GPR is not available. One can then evaluate a posteriori the concordance of fluxes with either proteomics-derived or transcriptomics-derived RAS, to identify more hierarchical regulatory layers, widening our understanding of metabolism. We remark that extension of the pipeline to cover allosteric and cofactor regulation would require explicit modeling of kinetic dynamics, which in turn requires knowledge of kinetic parameters. Alternatively, one might indirectly identify them by exclusion, that is, by supposing that fluxes that do not appear to be regulated by any of the other regulation levels are controlled by allosteric effects, or by looking for correlations between fluxes and metabolites within the same pathway or belonging to other cross-related biochemical pathways.

Knowing whether a flux is controlled at the metabolic or enzymatic level is mandatory in designing therapeutic strategies. If a putative therapeutic flux is controlled metabolically, direct targeting the corresponding metabolic enzyme will not produce any effect. On the contrary, identifying the metabolic reaction(s) that indirectly affect the target reaction allows for designing an effective therapeutic intervention. Hence, by integrating high-throughput omics data through mathematical models, INTEGRATE makes it possible to dissect the complex and intertwined regulation of metabolic networks and inform targeted strategies to counteract metabolic rewiring and/or dysfunction underlying different pathological disorders.

The pipeline, however, can be applied to any case study. By way of example, metabolic engineering efforts are already inspired by constraint-based modeling [52]. Expanding the toolbox of computational tools will surely increase the success rate of efforts toward the application of engineering concepts to living organisms, contributing to the development of predictable, scalable, and efficient biological devices, whose performance is not hampered by inadequate knowledge of the underlying design principles [53].

## Material and methods

### Experimental methods

**Cell culture.** MDA-MB231 and MCF7 cell lines were maintained and grown in Dulbecco's modified Eagle's medium (DMEM) containing 4 mM L-glutamine, supplemented with 10% fetal bovine serum (FBS). MDA-MB361 cell line was maintained and grown in DMEM/F12 containing 20% FBS and 4 mM L-glutamine. SKBR3 cell line was maintained and grown

in DMEM containing 2 mM L-glutamine, supplemented with 10% FBS. MCF102A cell line was maintained and grown in DMEM/F-12 containing 5% horse serum, 2.5 mM L-glutamine, 20 ng/ml EGF, 100 ng/ml cholera toxin, 0.01 mg/ml insulin, and 500 ng/ml hydrocortisone. All media were supplemented with 100 U/ml penicillin and 100 $\mu$g/ml streptomycin, and cells were incubated at 37˚C in a 5% $CO_2$ incubator. All reagents for media were purchased from Life Technologies (Carlsbad, CA, USA). Glucose concentration is 25 mM in DMEM medium and 17.5 mM in DMEM/F-12 medium.

**Cell proliferation and protein content analysis.** Cells were plated in 6-well cell culture plates (cat no. 657 160, Greiner Bio-One) in their respective growth medium (described above). In particular, 80.000 cells/well were seeded for MCF7 and MDA-MB231 cell lines, 120.000 cells/well for SKBR3 and MCF102A, 220.000 cells/well for MDA-MB361. Culture medium was replaced after 18 hours and cells were collected and counted after 0, 24, 48, and 72 hours. Cells pellets collected at indicated time points were resuspended in 100 ul of RIPA protein extraction buffer supplemented with protease inhibitor cocktail and placed on ice for 30 minutes. Samples were then centrifuged at 12.000 g for 10 minutes at 4˚C and the supernatant was collected. Protein content was evaluated through Bradford assay, using Pierce Coomassie Plus (Thermo Scientific). Coomassie was diluted 1:1 with water and 1ml of the solution was placed in a semi-micro cuvette along with 5ul of sample. The absorbance was read at 595 nm in a spectrophotometer (Cary 60 UV-VIS, Agilent Technologies). Protein concentration was determined using bovine serum albumin as a standard protein.

**Metabolite extraction from cell culture.** Cells were plated in 6-well plates with normal growth medium. The culture medium was replaced after 18 h, which corresponds to time 0 in the datasets used to constrain the computational models and then cells were incubated for 48 h. For metabolites extraction, cells were quickly rinsed with NaCl 0.9% and quenched with 500 $\mu$l ice-cold 70:30 acetonitrile-water. Plates were placed at -80˚C for 10 minutes, then the cells were collected by scraping, sonicated 5 seconds for 5 pulses at 70% power twice (the sonicator is a HD 2070 from Bandelin Sonoplus equipped with a MS73 probe) and then centrifuged at 12000 g for 10 min at 4˚C. The supernatant was collected in a glass insert and evaporated in a centrifugal vacuum concentrator (Concentrator plus/ Vacufuge plus, Eppendorf) at 30˚C for about 2.5 h using the V-AQ function, suitable for aqueous solutions. The speed is not adjustable (248 x g) as well as the pressure (pump maintains an ultimate pressure of at least 20 mbar. Its suction capacity is at least 1.8 m3/h). Samples were then resuspended with 150 $\mu$l of ultrapure water prior to analyses.

**LC-MS metabolic profiling.** LC separation was performed using an Agilent 1290 Infinity UHPLC system and an InfintyLab Poroshell 120 PFP column (2.1 x 100 mm, 2.7 $\mu$m; Agilent Technologies). The injection volume was 15 $\mu$L, the flow rate was 0.2 mL/min with column temperature set at 35˚C. Both mobile phase A (100% water) and B (100% acetonitrile) contained 0.1% formic acid. LC gradient conditions were: 0 min: 100% A; 2 min: 100% A; 4 min: 99% A; 10 min: 98% A; 11 min: 70% A; 15 min: 70% A; 16 min: 100% A with 5 min of post-run. Let us clarify that the method was optimized using two minutes in isocratic for better separation of the polar compounds that otherwise could coelute, and that the total run time per sample is 21 minutes, but the 5 minutes post-run were not acquired. MS detection was performed using an Agilent 6550 iFunnel Q-TOF mass spectrometer with Dual JetStream source operating in negative ionization mode. MS parameters were: gas temp: 285˚C; gas flow: 14 L/min; nebulizer pressure: 45 psig; sheath gas temp: 330˚C; sheath gas flow: 12 L/min; VCap: 3700 V; Fragment: 175 V; Skimmer: 65 V; Octopole RF: 750 V. Active reference mass correction was done through a second nebulizer using the reference solution (m/z 112.9855 and 1033.9881) dissolved in the mobile phase 2-propanol-acetonitrile-water (70:20:10 v/v). Data were acquired from m/z 60–1050. Data analysis and isotopic natural abundance

correction were performed with MassHunter ProFinder (Agilent). The Agilent Jet Stream ESI-MS source uses super-heated nitrogen as a gas. For further details on data processing the reader is referred to [54].

Raw data are deposited at www.ebi.ac.uk/metabolights/MTBLS3597. Normalized data (on protein *μg*) used for computational analyses are reported in S1 File.

**Metabolites quantification in the media samples.**   Absolute quantification of glucose, lactate, glutamine, and glutamate in fresh medium at t = 0 and in spent media after 48 hours of growth was determined enzymatically using YSI2950 bioanalyzer (YSI Incorporated, Yellow Springs, OH, USA). Media collected from experiments were thawed and centrifuged at 2000 rpm for 5 minutes prior to analysis. 1ml of sample is placed in a 1.5 ml tube and a volume of 25 ul was injected for the analysis. YSI bioanalyzer employed enzyme-based biosensors for measuring glucose, lactate, glutamate, and glutamine concentrations. The biosensors used oxidases-containing membranes for oxidizing substrates, releasing hydrogen peroxide. The hydrogen peroxide is detected amperometrically at a platinum electrode surface. The current flow at the electrode is directly proportional to the hydrogen peroxide concentration and hence to the substrate concentration. Glucose, lactate, glutamine, and glutamate standard solutions were used to calibrate the instrument.

**RNA extraction.**   Total RNA was extracted from at least 8 x 106 cells by using RNeasy Mini Kit (Qiagen), according to manufacturer's protocol. In detail, cells were disrupted by using lysis buffer, homogenized, and, finally, resuspended by adding 30 L of RNase-free water. Following RNA isolation, DNAse treatment was performed using DNAse I, RNase-free (ThermoFisher Scientific). After purification by ethanol precipitation, to assess the final RNA yield and purity, the spectrophotometer NanoDrop ND-1000 (NanoDrop Technologies) was employed by the means of 260/280 and 260/230 ratios. The 2200 TapeStation instrument (Agilent 741Technologies) also evaluated the RNA quality, in order to assess the RNA Integrity Number Equivalent (RIN) for each processed sample.

**RNA-Seq library preparation.**   RNA-Seq libraries were prepared using the Illumina TruSeq Stranded mRNA Library Prep Kit, according to manufacturer's instructions. For each of the five cell lines under study, three replicates were prepared. Each library was first analyzed on 2200 TapeStation instrument to check length and quality and then quantified by fluorescent dye PicoGreen (ThermoFisher Scientific) on NanoDrop ND-1000 to calculate the concentration.

RNA-Seq libraries were then diluted to 2nM concentration and normalized using standard library quantification and quality control procedures as recommended by the Illumina protocol. RNA-Seq libraries were sequenced using Illumina HiSeq2500 to obtain 150-bp paired-end reads. After fastq quality control by using FastQC tool, raw reads were mapped with STAR aligner (v.2.6.1d) to human reference genome (hg38) and gene counts were calculated by HTSeq (v.0.6.1), using the hg38 Encode-Gencode GTF file (v28) as gene annotation file. Gene abundance was measured in fragments per kb of exon per million fragments mapped (FPKM).

Raw reads are available in NCBI Short Reads Archive (SRA) under Accession Number PRJNA767228.

## ENGRO2 model reconstruction

Starting from ENGRO1 [44], we reconstructed a more extended and curated constraint-based core model of human metabolism. The reconstruction of the ENGRO2 model was based on a step-wise manual procedure using ENGRO1 model as a scaffold and progressively including specific pathways or reactions from Recon3D according to their relevance in literature for cancer cells.

The most invasive change that we implemented in the model was the compartmentalization of reactions and metabolites within the intracellular space. Many studies support the evidence of an altered expression of some mitochondrial carriers in multiple cancer cells that, most probably, arise as an adaptation to their current metabolic state and the consequent new requirements [55]. Therefore, in addition to the extracellular compartment, we divided the intracellular space into cytosol and mitochondrion, and we associated model reactions with each compartment according to literature knowledge. When needed, specific transport reactions were included to link biochemical transformations occurring within the cytosol and the mitochondrial matrix.

Another major extension of ENGRO1 to the ENGRO2 model concerned the addition of all reactions belonging to non-essential and essential amino acids metabolism.

We also included required cofactors within the stoichiometric equations of the model reactions. In fact, recent findings associated altered levels of some cofactors, including coenzyme A [56], water [57] and orthophosphate [58], with the emergence of various side effects and cancer promotion.

Moreover, to integrate gene expression data, we made all reactions belonging to the oxidative phosphorylation (OXPHOS) pathway explicit. A lumped version of the entire route was instead considered in the ENGRO1 model, in the form of two reactions representing the oxidation through the transfer of electrons from NADH and $FADH_2$ respectively. The stoichiometry of the complex I-like reaction also included the generation of reactive oxygen species (ROS), which are known to be generated by a deficient Complex I activity, following specific mutations affecting subunits of this enzymatic complex [59, 60]. In the ENGRO2 network, we included five separate reactions representing the reaction catalyzed by each OXPHOS complex. Because of ROS production from Complex I, it was necessary to add the ROS detoxification pathway by means of glutathione.

In view of the relevance of one-carbon metabolism in cancer cells [61], we enriched the ENGRO2 model by including both folate and the methionine cycles.

In ENGRO1, the oxidative branch of the pentose phosphate pathway was included only partially, that is, until the synthesis of the phosphoribosyl pyrophosphate (PRPP). PRPP is fundamental for cell biomass synthesis as the entry point of nucleotides biosynthesis. In view of the implications of one-carbon metabolism in purines and pyrimidines production, we extended the pentose phosphate pathway by including in ENGRO2 the complete nucleotides biosynthesis route. Moreover, we also included the non-oxidative branch of this pathway because of its relevance in reconverting its intermediates into glycolytic metabolites.

Recent evidence highlighted a crucial role of beta-oxidation in providing tumour cells with sources promoting growth and survival, especially under metabolic stress [62]. Moreover, fatty acid oxidation contributes to the cell total NADPH pool, by producing acetyl-CoA, which then enters the TCA cycle and is converted, together with oxaloacetate, to citrate. The high relevance of NADPH is linked to its role in providing redox power for tumor cells to counteract oxidative stress. For these reasons, we included the mitochondrial beta-oxidation pathway in ENGRO2 to degrade fatty acids, which in the model are represented under the form of palmitate.

Another recent finding that we considered during the reconstruction of ENGRO2 regards the dysregulation of polyamines metabolism and its requirement under the neoplastic condition [63]. In view of this evidence, we included reactions belonging to the polyamines metabolism.

We defined the biomass synthesis reaction exactly as in Recon3D model, in terms of set of metabolites considered and corresponding stoichiometric coefficients. Given that ENGRO2 does not explicitly include lipid synthesis, we assigned the sum of the stoichiometric

coefficients of 1-Phosphatidyl-1D-Myo-Inositol, Phosphatidylcholine, Phosphatidylethanol-amine, Phosphatidylglycerol, Cardiolipin, Phosphatidylserine, and Sphingomyelin to palmitate.

In order to assign Gene-Protein-Reaction associations to all model reactions, when possible, we relied on the HMR core model [64], where GPRs were manually refined [39]. For remaining reactions, we relied on the Recon3D model.

The final list of reactions included in ENGRO2, their GPRs, and indication on the pathway they belong to, is reported in S3 File.

### Reaction Activity Score computation

Catalysis of a reaction by a given set of enzymes is encoded within the model through gene-protein-reaction (GPR) rules [27–29]. GPR rules are logical expressions that exploit the AND and OR logical operators to describe the different types of relationships that can exist among enzymes. In particular, the AND operator is used when distinct genes encode multiple subunits of the same enzyme, implying that all the subunits are equally necessary for the reaction to take place. On the contrary, the OR operator is used when distinct genes encode multiple isoforms of the same enzyme, entailing that either isoform is enough for the reaction to be catalyzed. These logical operators can be combined to describe more complex scenarios involving both isoforms and subunits.

As commonly done (e.g., in [39, 49, 65]), we combined the RNA-seq datasets with the GPR rules through the employment of the Reaction Activity Score (RAS). The RAS is based on the assumption that enzyme isoforms contribute additively to the reaction activity, whereas the least expressed enzyme subunit is limiting. For each cell line $c$ in the set $C$ of cell lines under study, for each sample $\xi$, and for each reaction $r$, we computed the corresponding $\text{RAS}^c_{r,\xi}$, by resolving the corresponding logical expression as follows. When multiple genes are joined by an AND operator, we took the minimum transcript level value. When multiple genes are joined by an OR operator, we took the sum of their values. More in detail, the RAS related to reactions with the AND operator was computed using the following formula:

$$\text{RAS}^c_{r,\xi} = \min\{T_{g,c} : g \in G_r\} \tag{1}$$

where $G_r$ is the set of genes that encode the subunits of the enzyme catalyzing reaction $r$. The RAS related to reactions with the OR operator is computed as:

$$\text{RAS}^c_{r,\xi} = \sum T_{g,c} : g \in G_r. \tag{2}$$

In case of GPRs combining both operators, we respected their standard precedence. We then computed the reaction activity score of a cell line, by averaging over the samples as $\text{RAS}^c_r = \langle \text{RAS}^c_{r,\xi} \rangle_\xi$. Once that $\text{RAS}^c_r$ of each reaction was computed for each investigated cell line, they were normalized on the maximum $\text{RAS}^c_r$ of all cell lines:

$$\overline{\text{RAS}}^c_r = \frac{\text{RAS}^c_r}{\max\{\text{RAS}^c_r\}_c}. \tag{3}$$

When the RAS was equal to 0 in all the cell lines, the corresponding normalized $\overline{\text{RAS}}$ was kept equal to 0. For reactions not associated with a GPR the RAS was set to 1.

### From ENGRO2 to cell-relative constraint-based models

In order to predict the differences in the fluxes of the five cell lines under study, we relied on constraint-based modeling. Constraint-based modeling is based on a steady state assumption

for internal metabolites. Given a $M \times N$ stoichiometric matrix S, with M metabolites and N reactions, and a flux vector $\boldsymbol{v}$, the steady state assumption imposes that $S \cdot \boldsymbol{v} = 0$. Hence, the null space of matrix $S$ is the space of flux vectors consistent with the steady state assumption. To mimic as closely as possible the biological process in analysis, it is possible to bound the space of feasible fluxes by means of convex half-planes represented by the vectors $\boldsymbol{v_L}$ and $\boldsymbol{v_U}$, which specify, respectively, the lower and upper bounds for the components of the flux vector $\boldsymbol{v}$ [66].

In order to differentiate the five models, we set flux boundaries to the generic reconstruction of human metabolism ENGRO2, according to the differences observed in the experimental datasets. We distinguish three families of constraints that we detail in the following. We remark that even if the three types of constraints are incremental (meaning that type 2 cannot be applied after type 3) one can omit type 2 constraints and apply type 3 directly after type 1.

**Constraints on nutrient availability—Type 1.** This set of constraints on exchange fluxes define the cell growth medium. We defined the set of nutrients that can be internalized by each of the five investigated cell lines according to the composition of the two exploited experimental mediums. For every uptaken metabolite, an exchange reaction was included within the network by setting its upper bound to 0 and tuning its lower bound proportionally to the corresponding concentration contained in the corresponding growth medium.

**Constraints on extracellular fluxes—Type 2.** This additional set of constraints on exchange fluxes defines the ratio between consumption and secretion of nutrients. We set this type of constraints according to the experimentally determined flux ratios of lactate to glucose, lactate to glutamine, and glutamate to glutamine. In detail, we proceeded as follows:

- We took the concentration values of the two nutrients, glucose and glutamine, and of the two byproducts, glutamate and lactate, that we previously obtained with the YSI analyzer of spent medium.

- To derive the amount either consumed or produced of each metabolite mentioned above, we computed the difference between its concentration at the initial time (time 0) and after 48 hours of growth. We remark that we intend to estimate ratios between consumption and secretion of metabolites, rather than consumption or secretion rates. Hence, it is irrelevant to divide the difference between concentrations at different time points by the integral of cells number.

- We computed the lactate produced over glucose consumed, lactate produced over glutamine consumed, and glutamate produced over glutamine consumed ratios for each sample. We then considered the two biological replicates separately, while averaging over the ratios obtained for the three technical replicas of each biological replicate. The specific values are reported in S1 File.

- We added the obtained flux ratios as further constraints to the model. To give an example, the lactate to glucose ratio was translated into a model constraint using the following expression:

$$-\sigma_{\text{Lac/Glc}} \quad \leq \quad v_{\text{ExLac}} - \overline{x}_{\text{Lac/Glc}} \cdot v_{\text{ExGlc}} \quad \leq \quad \sigma_{\text{Lac/Glc}}, \tag{4}$$

where the ratio of the lactate secretion flux $v_{\text{ExLac}}$ over the glucose consumption flux $v_{\text{ExGlc}}$ ranges between minus one and plus one standard deviation $\sigma_{Lac/Glc}$ of the mean lactate to glucose ratio $\bar{x}_{\text{Lac/Glc}}$ of the two biological replicas.

**Trascriptomics-derived constraints—Type 3.** All constraints on external fluxes being equal, different cells might still exploit different flux distributions for internal reactions. We

assume that a cell with lower enzyme activity for a given internal reaction $r$ has in principle a lower capability to carry flux through such reaction. Hence, we let the flux boundaries $v_L$ and $v_U$ of a given cell $c$ depend on the $\overline{RAS}_r^c$ score.

In order to properly limit the flux of an internal reaction $r$ based on its RAS relative score, as a preliminary step, the maximum and minimum flux through such reaction, based on extracellular flux constraints (i.e., type 1 and type 2 constraints), must be determined specifically for each cell and reaction. In fact, limiting the generic upper bounds $v_U$ might have no effect if the actual flux capacity $v_{U,r}^c$ of reaction $r$ in cell $c$ ($v_{L,r}^c$ for backward reactions) is lower than the former. To this aim, we performed a Flux Variability Analysis (FVA). FVA [67, 68] is a constraint-based modelling technique aimed at determining the maximal (and minimal) possible flux through any reaction of the model and thus evaluating the cell's range of metabolic capabilities, given a set of constraints. FVA solves the following two linear programming optimization problems (one for minimization and one for maximization) for each flux $v_i$ of interest, with $i = 1, \ldots, N$:

$$
\begin{aligned}
&max/min \ v_i \\
&\text{subject to} \ S \cdot \boldsymbol{v} = 0 \\
&\boldsymbol{v}_L \le \boldsymbol{v} \le \boldsymbol{v}_U
\end{aligned} \tag{5}
$$

FVA has been largely used to determine the variability of fluxes within the set of (sub)optimal solutions. On the contrary, we exploited it to determine the variability of fluxes within the set of solutions satisfying type 1 and type 2 constraints. Even if we did not ask the network to produce biomass (sub)optimally, we still wanted to exclude completely unrealistic flux distributions. Therefore, we constrained the order of magnitude of the growth yield on glucose to be within the minimum yield ($3.90762 \times 10^{-5}$) and maximum yield ($1.67998 \times 10^{-4}$) that were observed experimentally across all samples. Specifically, we set the constraint on growth yield for all the five cell lines as:

$$
min_{Yield} \cdot v_{ExGlc} \cdot mw_{Glc} \cdot 0.001 \le 0.131972 \cdot \ v_{Biomass} \le max_{Yield} \cdot v_{ExGlc} \cdot mw_{Glc} \cdot 0.001, \tag{6}
$$

where $v_{Biomass}$ is the biomass synthesis flux, $v_{ExGlc}$ is the glucose uptake flux, $mw_{Glc}$ is the molecular weight of glucose, $min_{Yield}$ and $max_{Yield}$ are the minimum and maximum of the growth yield, and 0.131972 is the fraction of proteins within biomass composition.

Once we obtained the vectors $\boldsymbol{v}_L{}^c$, $\boldsymbol{v}_U{}^c$ through FVA, we set the RAS-dependent flux boundaries of a GPR-associated reaction $r$ in cell $c$ as follows:

$$
\overline{RAS}_r^c \cdot v_{L,r}^c \le v_r^c \le \overline{RAS}_r^c \cdot v_{U,r}^c. \tag{7}
$$

Otherwise, if $r$ is not associated with a GPR, we set its boundary simply as:

$$
v_{L,r}^c \le v_r^c \le \cdot v_{U,r}^c. \tag{8}
$$

It is worth mentioning that we used Eq 8 also for two reactions associated with a GPR, namely CARPEPT1tc and HIStiDF, because their null RAS were preventing growth in some of the cell lines. The RAS score of CARPEPT1tc reaction is null in MDAMB231 and MCF7 cell lines, whereas the RAS score of HIStiDF reaction is null in MCF7 cell line.

## Generation of FFD dataset, via random sampling

Once the cell-relative models were created, for all downstream analyses described hereinafter, we converted them into irreversible models, in which reversible reactions are represented with two distinct and complementary forward reactions. We remark that this step must be

performed necessarily after and not before defining type 2 constraints, to avoid basing them on unrealistically high FVA values caused by futile loops.

Uniform sampling of the constrained null space of $S$ is a powerful tool to compare metabolism under different conditions [69, 70]. In this work, we exploited the implementation of optGpSampler algorithm [71] available in COBRApy [72], and we sampled a million steady state solutions of the ENGRO2 model in all the tested conditions. To get a large number of samples, we used the batch generator option of the algorithm, creating ten batches of 100.000 samples each. We set the other parameters to the default value, except for the thinning value that we set to 10.

Let us specify that the constraint on the growth yield on glucose in Eq 6 is retained during this sampling step.

## Generation of RPS dataset

Let $x^i$ be the vector of abundances of the chemical species in a given steady state $i$ of the metabolic network. Let $v^i$ be the vector of reaction flux rates in the same steady state $i$. The flux of chemical species through a reaction is the rate of the forward reaction, minus that of the reverse reaction (in molecules per unit of time). When dealing with irreversible models, the reaction flux and the rate coincide.

The following assumptions allow one to analytically estimate relative fluxes from relative abundances:

- for each reaction $r$ in the system, the mass action law is assumed: $v_r = k_r \ \prod_{q=1}^{N} [X_q]^{s_{r,q}}$, where $k_r$ is the kinetic constant of reaction $r$ and $X_q$ is the $q^{th}$ substrate of the $N$ total substrates of reaction $r$, and $s_{r,q}$ is the stoichiometric coefficient of substrate $X_q$ in reaction $r$ i.e., how many molecules of the substrate partake to the reaction;

- the kinetic constant $k_r$ of a given reaction $r$ is assumed to not vary between two steady states $i$ and $j$.

Given such assumptions, the variation between the flux of an irreversible reaction $r$ in two steady states $i$ and $j$ can thus be computed as the ratio $v_r^i / v_r^j$:

$$\frac{v_r^i}{v_r^j} = \prod_{q=1}^{N} \left( \frac{[X_q]^i}{[X_q]^j} \right)^{s_{r,q}}, \tag{9}$$

which does not depend on $k_r$.

It goes without saying that if the numerator is higher than the denominator, then flux $v_r$ in steady state $i$ is higher than flux $v_r$ in steady state $j$. Therefore, in order to compare the substrate contribution to the reaction rate in different cell lines, we computed for each reaction $r$ and for each cell line $c$ (assumed at steady state) a Reaction Propensity Score (RPS), defined as follows:

$$\text{RPS}_r^c = \prod_{q=1}^{N} ([X_q])^{s_{r,q}}. \tag{10}$$

Of course, the above assumptions does not often hold. However, the $\text{RPS}_r$ allows us to predict whether the flux of a reaction in a cell line is expected to be different as compared to another cell line, just considering substrate availability. Nothing prevents the observed flux variation to be consistent with the RPS variation, even if the enzyme activity actually differs between the two cell lines under study.

## Concordance analysis

Once the FFD, RAS, and RPS values were obtained for each cell line $c$, we wanted to assess whether the sign of their variation between cell lines, for any given reaction $r$, was consistent with respect to one another.

To assess the sign of the variation of FFD values, i.e. up (+1), down (-1), or no-change (0), we first performed the Mann-Whitney U test [73] (p-value $< 0.05$) between the FFD distributions of each pair of the five cell lines. In parallel, the $\log_2$ of the ratio between the median flux of the two cell lines was also computed.

Concerning the RAS and the RPS values, we first performed a t-test (p-value $< 0.05$) between the scores of the samples of each pair of the five cell lines. We also computed the $\log_2$ of the ratio between the average scores of the two cell lines.

Finally, we registered the sign of the variation for each pair of the five cell lines under study (for a total of 10 pairs) according to each of the three datasets. At first instance, a positive sign was registered if the distribution of samples values of the first member of the comparison was statistically higher (according to the statistical tests described above) and if the average or median value was at least 20% higher. A negative sign was registered if the distribution of samples values of the first member of the comparison is statistically lower and if the average or median value was at least 20% lower. A 0 was registered otherwise. We used a relaxed threshold for the fold-change as in [39] because even a difference of 20% in genes encoding members of a metabolic pathway may dramatically alter the flux through the pathway. Yet, this parameter can be modified arbitrarily.

We quantified the level of concordance of the 10 variation signs (1 for each pair of cell lines) for a given pair of datasets by means of the Cohen's kappa metric, which has been commonly used to measure inter-rater reliability for qualitative (categorical) items [74].

A toy example recapitulating all the steps to obtain the concordance scores, including RAS and RPS computation, is illustrated in S4 Fig.

To obtain the empirical probability of agreement between two independent datasets, we randomly sampled 1000 times (with replacement) the distribution of RPS pairwise variations. We sampled the results of the statistical tests (-1,0,+1), rather than randomly sampling the original RPS values and then performing the tests, to avoid obtaining an over-representation of zeros (i.e., rejections of the null hypothesis of the t-test). We then computed Cohen's kappa values between the sampled RPS variations and the original FFDs variations for each pair of cell lines. We then used the empirical distribution of the obtained Cohen's kappa values to associate a p-value to each reaction. Finally, we adjusted the p-value with the Benjamini and Hochberg procedure to keep the false discovery rate (FDR) below 5%.

## Supporting information

**S1 Fig. Types of reaction flux controls.** Graphical explanation of transcriptional, metabolic and combined metabolic and transcriptional control.
(PDF)

**S2 Fig. ENGRO2 map—Part I.** Graphical representations of central carbon metabolism reactions included in ENGRO2 model.
(PDF)

**S3 Fig. ENGRO2 map—Part II.** Graphical representations of essential amino acid metabolism reactions included in ENGRO2 model.
(PDF)

**S4 Fig. Toy example.** Diagram recapitulating all the INTEGRATE steps for a specific toy reaction.
(PDF)

**S1 File. Input experimental datasets.** Read counts (FPKM), LC-MS metabolic profiling, extracellular fluxes and protein content.
(XLSX)

**S2 File. ENGRO2 model.** SBML of the unconstrained model.
(XML)

**S3 File. ENGRO2 model.** XLSX of the unconstrained model.
(XLSX)

**S4 File. Concordance scores of ENGRO2.** Table of RPSvsRAS and RPSvsFFD Cohen's kappa coefficients, empirical and adjusted p-values and confidence intervals for all model reactions (when applicable).
(XLSX)

**S5 File. Concordance scores of Recon3D.** Table of RPSvsRAS and RPSvsFFD Cohen's kappa coefficients confidence intervals for all model reactions (when applicable).
(XLSX)

**S1 Compressed File Archive. Cell relative metabolic models.** Compressed file archive including the five cell-relative models (SBML).
(ZIP)

## Acknowledgments

We warmly thank Alex Graudenzi, Daniele Ramazzotti, and Davide Maspero for useful suggestions.

## Author Contributions

**Conceptualization:** Lilia Alberghina, Marco Vanoni, Chiara Damiani.

**Data curation:** Marzia Di Filippo, Bruno Giovanni Galuzzi.

**Formal analysis:** Marzia Di Filippo, Dario Pescini, Bruno Giovanni Galuzzi, Chiara Damiani.

**Funding acquisition:** Lilia Alberghina, Marco Vanoni.

**Investigation:** Marzia Di Filippo, Dario Pescini, Bruno Giovanni Galuzzi, Marcella Bonanomi, Daniela Gaglio, Eleonora Mangano, Clarissa Consolandi, Chiara Damiani.

**Methodology:** Marzia Di Filippo, Dario Pescini, Bruno Giovanni Galuzzi, Chiara Damiani.

**Software:** Marzia Di Filippo, Dario Pescini, Bruno Giovanni Galuzzi, Chiara Damiani.

**Supervision:** Chiara Damiani.

**Visualization:** Marzia Di Filippo, Dario Pescini, Bruno Giovanni Galuzzi, Chiara Damiani.

**Writing – original draft:** Marzia Di Filippo, Chiara Damiani.

**Writing – review & editing:** Dario Pescini, Lilia Alberghina, Marco Vanoni.

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
