## [Decision Letter · Decision Letter 0]

27 Sep 2021

Dear Dr Damiani,

Thank you very much for submitting your manuscript "INTEGRATE: Model-based multi-omics data integration to characterize multi-level metabolic regulation" for consideration at PLOS Computational Biology.

As with all papers reviewed by the journal, your manuscript was reviewed by members of the editorial board and by several independent reviewers. In light of the reviews (below this email), we would like to invite the resubmission of a significantly-revised version that takes into account the reviewers' comments.

We cannot make any decision about publication until we have seen the revised manuscript and your response to the reviewers' comments. Your revised manuscript is also likely to be sent to reviewers for further evaluation.

Sincerely,

Daniel Machado

Guest Editor

PLOS Computational Biology

Kiran Patil

Deputy Editor

PLOS Computational Biology

Reviewer's Responses to Questions

**Comments to the Authors:**

Reviewer #1: Uploaded as attachement

Reviewer #2: Review is uploaded as an attachment.

**Have the authors made all data and (if applicable) computational code underlying the findings in their manuscript fully available?**

Reviewer #1: **No: **I was not able to review the code in the GitHub repository (https://github.com/qLSLab/integrate) as it was not public.

Reviewer #2: **No: **Experimental data is missing to some extend, the provided github is not working.

PLOS authors have the option to publish the peer review history of their article (what does this mean?). If published, this will include your full peer review and any attached files.

Reviewer #1: No

Reviewer #2: No
---

## [Decision Letter · Decision Letter 1]

3 Jan 2022

Dear Dr. Damiani,

Thank you very much for submitting your manuscript "INTEGRATE: Model-based multi-omics data integration to characterize multi-level metabolic regulation" for consideration at PLOS Computational Biology. As with all papers reviewed by the journal, your manuscript was reviewed by members of the editorial board and by several independent reviewers. The reviewers appreciated the attention to an important topic. Based on the reviews, we are likely to accept this manuscript for publication, providing that you modify the manuscript according to the review recommendations.

Sincerely,

Daniel Machado

Guest Editor

PLOS Computational Biology

Kiran Patil

Deputy Editor

PLOS Computational Biology

[LINK]

Reviewer's Responses to Questions

**Comments to the Authors:**

Reviewer #1: I'm happy to see that the authors have made a true effort to address most of the shortcomings that were suggested previously, and I believe that the manuscript has improved a lot. The minor revisions requested below are mostly text or figure edits that in my opinion will improve the manuscript.

Comments (line numbers refer to manuscript with changes higlighted)

- Line 64: Do you mean "a significant increase the abundance of enzyme E"? An increase in activity of enzyme E is by definition an increased flux the reaction r (Enzyme activity = moles of substrate converted per unit time).

- Line 77: .. when the amount of enzyme E increases.

- I think the examples of control mechanisms (line 76-86) can be more accurately explained, possibly by illustratin how the Michelis-Menten equation can be approximated when substrate concentration is << or >> than Km.

- Line 79: The description of metabolic control doesn't make sense. Increased enzyme and substrate abundance have to increase flux. I think it is more correct to say the rate is mostly affected by substrate concentration. If you consider the Michelis-Menten equation, enzyme abundance also affects reaction rate when S<<km.>- Line 123-124: This sentence should be rephrased.

- Line 133: typo: starting form

- Line 153: typo: A a proof ..

- Figure 2D: Maybe I'm misinterpreting the meaning of the size of each dot, but in my understanding it is the ratio of samples for each cell line for each metabolite above the average value for that metabolite across all cell lines. If this is correct, how can all the cell lines have large (close to 100%) of samples above the mean for some metabolites?

- Figure 2F: Y-axis numbers too small

- Line 244-245: Seems like a word is missing here

- Fig 3: Please report p-values for the calculated correlations

- Line 425: missing parenthesis

- Figure 5A, the two heatmaps miss titles (which is RPS and which is FFD)

- Figure 5: A proper legend for colors/cell lines would be nice

- GitHub repository should be deposited at a permanent repository (e.g. Zenodo) in agreement with FAIR principles.</km.>

Reviewer #2: I would like to thank the authors for their contribution. I hope that they agree that the comments from me and the other Reviewer increased the readibility and value of the scientific contribution to the field.

I think there are still some minor improvements that could be done (language, sentence length,...) but over all I think the manuscript is good for publication.

**Have the authors made all data and (if applicable) computational code underlying the findings in their manuscript fully available?**

Reviewer #1: Yes

Reviewer #2: Yes

PLOS authors have the option to publish the peer review history of their article (what does this mean?). If published, this will include your full peer review and any attached files.

Reviewer #1: No

Reviewer #2: No

Figure Files:

Data Requirements:

Reproducibility:

References:

---

## [Editor Report · Decision Letter 2]

13 Jan 2022

Dear Dr. Damiani,

We are pleased to inform you that your manuscript 'INTEGRATE: Model-based multi-omics data integration to characterize multi-level metabolic regulation' has been provisionally accepted for publication in PLOS Computational Biology.

Best regards,

Daniel Machado

Guest Editor

PLOS Computational Biology

Kiran Patil

Deputy Editor

PLOS Computational Biology

---

## [Editor Report · Acceptance letter]

31 Jan 2022

PCOMPBIOL-D-21-01369R2 

INTEGRATE: Model-based multi-omics data integration to characterize multi-level metabolic regulation

Dear Dr Damiani,

I am pleased to inform you that your manuscript has been formally accepted for publication in PLOS Computational Biology. Your manuscript is now with our production department and you will be notified of the publication date in due course.

With kind regards,

Anita Estes
